# ReVISE: Learning to Refine at Test-Time via Intrinsic Self-Verification

Hyunseok Lee [* 1]  Seunghyuk Oh [* 1]  Jaehyung Kim [2]  Jinwoo Shin [1]  Jihoon Tack [1]

## Abstract

Self-awareness, i.e., the ability to assess and correct one's generation, is a fundamental aspect of human intelligence, making its replication in large language models (LLMs) an important yet challenging task. Previous works tackle this by employing extensive reinforcement learning or relying on large external verifiers. In this work, we propose Refine via Intrinsic Self-Verification (ReVISE), an efficient and effective framework that enables LLMs to self-correct their outputs through self-verification. The core idea of ReVISE is to enable LLMs to verify their reasoning processes and continually rethink reasoning trajectories based on its verification. To implement this efficiently, we introduce a structured curriculum based on preference learning. Specifically, as ReVISE involves two challenging tasks (i.e., self-verification and reasoning correction), we tackle each task sequentially using curriculum learning, collecting both failed and successful reasoning paths to construct preference pairs for efficient training. During inference, our approach enjoys natural test-time scaling by integrating self-verification and correction capabilities, further enhanced by our proposed confidence-aware decoding mechanism. Our experiments on various reasoning tasks demonstrate that ReVISE achieves efficient self-correction and significantly improves the reasoning performance of LLMs.

## 1. Introduction

Large language models (LLMs) have demonstrated remarkable success across diverse domains, such as coding assistants (Zhang et al., 2024b), search engines (Xiong et al., 2024), and personal AI assistants (Sajja et al., 2024), progressively advancing toward human-like logical reasoning capabilities (Amirizaniani et al., 2024). However, tasks requiring rigorous System 2 thinking—such as complex reasoning (Jaech et al., 2024), iterative trial-and-error (Song et al., 2024), and dynamic planning (Xie & Zou, 2024)—remain highly challenging (Lowe, 2024; Cai et al., 2024). A key difficulty in LLM reasoning is that errors in early steps can accumulate over time, leading to substantial inaccuracies (LeCun, 2022), while the models' intrinsic ability to detect and rectify such self-generated errors—often framed as a form of self-awareness—remains insufficient. This issue is further exacerbated by the autoregressive nature of LLMs, which constrains their ability to revisit and revise prior steps (Bachmann & Nagarajan, 2024).

To tackle this issue, recent approaches have emphasized verification (or correction) of LLM-generated reasoning trajectories as a crucial mechanism (Zhang et al., 2024a; Madaan et al., 2023). For instance, some methods utilize external large-scale verifiers to iteratively validate outputs and trigger regeneration (Luo et al., 2024). However, the reliance on expensive external models introduces computational inefficiencies. Alternatively, reinforcement learning (RL)-based techniques have shown promise in improving reasoning accuracy by optimizing reward signals based on ground-truth correctness, enabling self-correction (Kumar et al., 2024). However, RL is a complex and often unstable procedure (Mnih et al., 2015; Rafailov et al., 2023), and it does not explicitly model the verification of intermediate reasoning steps, making it difficult to assess whether a model is confident in its current trajectory or prone to deviating toward incorrect conclusions, which may limit interpretability and adaptability in complex reasoning tasks.

This raises a key question: *Can LLMs be equipped with an internal mechanism to explicitly verify their own reasoning and correct potential errors based on their verification?*

**Contribution.** We propose **Re**fine **V**ia **I**ntrinsic **SE**lf-Verification (ReVISE)[1], a novel and effective self-correction framework for LLM reasoning using self-verification. The core idea of ReVISE is to enable LLMs to assess their reasoning process and refine reasoning trajectories based on self-verification. Specifically, we introduce a special token, which outputs whether to stop the generation or revise the reasoning trajectory. To train the model to utilize this token

---

*Equal contribution  [1]KAIST  [2]Yonsei University. Correspondence to: Jihoon Tack <jihoontack@kaist.ac.kr>.

*Proceedings of the $42^{nd}$ International Conference on Machine Learning*, Vancouver, Canada. PMLR 267, 2025. Copyright 2025 by the author(s).

[1]Code available at: github.com/seunghyukoh/revise

effectively, we design a two-stage curriculum to simplify the learning of two challenging tasks—self-verification and self-correction—by breaking them into separate training stages. Here, both stages employ preference learning, allowing the model to learn these tasks efficiently without heavy computational overhead. In the first stage, we collect pairs of correct and incorrect reasoning trajectories (i.e., positive and negative samples for preference learning) based on output correctness to develop the model's self-verification ability. In the second stage, we generate new preference pairs for self-correction by constructing positive samples where a correct reasoning path follows an incorrect one, and negative samples where an incorrect reasoning path follows a correct one.

Furthermore, we introduce an inference-time scaling strategy for ReVISE that leverages self-verification to enhance performance. First, as ReVISE inherently verifies and refines reasoning paths when it detects incorrect outputs, it naturally benefits from increased test-time computation. Additionally, we propose a novel test-time sampling scheme that incorporates self-verification confidence (i.e., the confidence in deciding whether to terminate generation). Specifically, we integrate this confidence into existing test-time sampling methods by adjusting the sampling score based on the predicted confidence, leading to more reliable output.

We demonstrated the effectiveness of ReVISE through evaluations on multiple reasoning datasets across mathematical and coding domains. Notably, ReVISE enhances reasoning performance beyond prior methods, improving accuracy from 27.1→31.1% on GSM8K (Maj@3) (Cobbe et al., 2021) with Llama3 1B (Dubey et al., 2024) and from 33.2→36.0% on MATH (Maj@3) (Hendrycks et al., 2021) with Llama3 8B. Furthermore, our experimental results show that ReVISE consistently improves accuracy without relying on external feedback mechanisms, which often degrade performance on complex reasoning tasks. For instance, unlike approaches such as Refine (Madaan et al., 2023), which struggle when combined with existing models on complex tasks, ReVISE achieves these gains purely through self-verification and self-correction. Finally, we show that the proposed sampling scheme is more efficient than other sampling strategies when applied to models trained with ReVISE, further enhancing the performance.

## 2. Related Work

**LLM reasoning.** LLMs have made significant progress in reasoning through techniques such as Chain-of-Thought (CoT) prompting, fine-tuning, and self-improvement. CoT prompting, introduced by (Wei et al., 2022) and expanded by (Kojima et al., 2022) enables models to break down complex problems into intermediate steps, improving performance and interpretability. Structured reasoning meth-

ods, including self-consistency (Wang et al., 2022) and Tree-of-Thought (ToT) (Yao et al., 2024), enhance multi-step problem-solving by exploring various reasoning paths. Huang et al. (2022) have demonstrated self-improvement through iterative feedback, refining their outputs over time. Ensuring the reliability of reasoning approaches such as Reflexion (Shinn et al., 2024) and Self-Refine (Madaan et al., 2023) introduce iterative feedback loops, while verification techniques like step-by-step validation (Lightman et al., 2023) help maintain consistency and reduce errors. Unlike prior approaches, ReVISE learns self-verification during training, reducing train-test discrepancy and enabling more natural verification at inference.

**Test-time scaling for LLMs.** Recent works explored that scaling test-time computation, such as best-of-N sampling, can be even better than scaling train-time computation for performance (Snell et al., 2024). Specifically, test-time scaling strategies improve LLM performance by generating numerous candidate outputs and selecting the best. To enhance decision-making, external verifiers are often employed to evaluate and refine these outputs (Liang et al., 2024). Moreover, Kumar et al. (2024); Qu et al. (2024) applied extensive reinforcement learning to overcome the efficiencies and dependence on the verifier's performance. In safety research, backtracking methods have introduced reset tokens to correct unsafe responses (Zhang et al., 2024c). While they focus on reducing the likelihood of unsafe outputs with limited second attempts to refuse answers, our approach targets complex reasoning tasks enabled by self-correction through an explicit verification process and two-stage curricula.

**Self-improvement for LLMs.** Self-training methods enable LLMs to refine themselves using their own outputs. Supervised fine-tuning (SFT) (Brown et al., 2020b) trains on human-annotated data but lacks self-correction (Huang et al., 2023). Rejection fine-tuning (RFT) (Yuan et al., 2023) improves robustness by filtering low-quality responses but discards useful learning signals. STaR (Zelikman et al., 2022) iteratively fine-tunes models on self-generated solutions but struggles with compounding errors due to the absence of explicit verification. V-STaR (Hosseini et al., 2024) extends STaR by jointly training a verifier alongside the generator, leveraging both correct and incorrect responses to improve self-assessment, though it still depends on large-scale self-generated data. However, discovering high-quality solutions remains a challenge, as (Luong et al., 2024) shows that RL-based fine-tuning is ineffective without supervised initialization. Kim et al. (2024) explore using a stronger LLM to refine incorrect rationales from a smaller model, though Huang et al. (2024) argue that LLMs struggle with self-correction. Our approach integrates both generation and verification, leveraging correct and incorrect responses for more effective self-improvement.

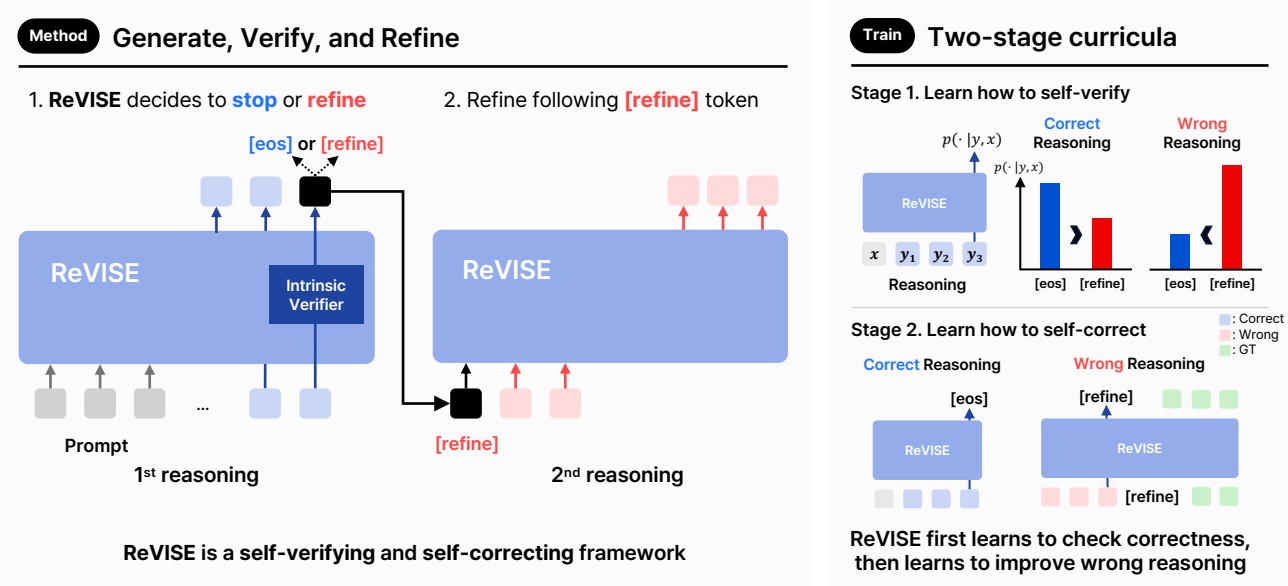

*Figure 1.* **Overview of ReVISE. Left:** ReVISE is a self-verifying and self-correcting reasoning framework. It first generates an initial answer, verifies its correctness, and decides whether to stop or refine. If the model generates the [refine] token, it refines the initial reasoning. **Right:** The structured curriculum-based training pipeline of ReVISE. In the first stage, the model learns self-verification by selecting between [eos] and [refine]. In the second stage, it learns to correct reasoning mistakes using golden data.

## 3. Learning to Refine at Test-Time via Intrinsic Self-Verification

In this section, we present Refine via Intrinsic Self-Verification (ReVISE), an LLM reasoning framework that self-verifies and refines the reasoning trajectory based on the verification. We first introduce the problem of interest and a special token coined [refine], which is used for refining the LLM's generation (in Section 3.1). Then, we present the core training method, namely the two-stage curricula (in Section 3.2) and the test-time inference strategy (in Section 3.3). The overview of ReVISE is depicted in Figure 1.

### 3.1. Problem setup: Learning to Verify and Refine

We describe the problem setup of our interest, i.e., self-verification and refinement. Given an input $x$, the initial output $y_{\texttt{init}}$ is sampled from the LLM $\mathcal{M}$, i.e., $y_{\texttt{init}} \sim \mathcal{M}(\cdot|x)$, where the reasoning path is included in $y_{\texttt{init}}$. The goal is to train an LLM that verifies the correctness of $y_{\texttt{init}}$ and decides whether to terminate generation or continue generating by refining its reasoning. To this end, we introduce a special token [refine] that determines whether to proceed with refinement. Specifically, given $y_{\texttt{init}}$, the model verifies its correctness by predicting $v \sim \mathcal{M}(\cdot|y_{\texttt{init}}, x)$, where $v \in \{[\texttt{eos}], [\texttt{refine}]\}$, allowing it to either terminate generation by predicting [eos] or continue generating by refining its reasoning by outputting [refine]. If refinement is needed, the model generates a revised response

$y_{\texttt{refined}} \sim \mathcal{M}(\cdot|[\texttt{refine}], y_{\texttt{init}}, x)$, completing the correction cycle. Note that this modeling has distinct advantages as one can access the model's verification confidence of $v$.

### 3.2. ReVISE: Refine via Intrinsic Self-Verification

We first describe our core training pipeline of ReVISE, namely the structured curriculum based on online preference learning. As ReVISE involves two challenging tasks (i.e., self-verification and refinement), we propose two-stage curricula. In the first stage, we train the LLM to intrinsically self-verify its generation by predicting the [eos] or [refine] tokens. Then, at the second stage, we continually train this LLM to correct the generation when the output reasoning is wrong. For efficient and stable training, we employ preference optimization (i.e., learning from preference-based positive and negative pairs) based on our proposed preference data collection strategy. This allows us to perform structured preference learning without relying on reinforcement learning (RL), which can be computationally extensive and unstable (Rafailov et al., 2023).

**Stage 1: Learning to verify self-generations.** Given an initial LLM $\mathcal{M}_0$ and a supervised fine-tuning dataset $\mathcal{D} = \{(x_i, y_i)\}_i$ consisting of input-label pairs (including reasoning traces), our goal is to construct preference pairs for training $\mathcal{M}_0$. Specifically, for each input $x$, we generate a positive output $y^+$ and a negative output $y^-$. To achieve this, we first sample multiple responses from $\mathcal{M}_0$. This

allows us to obtain both correct reasoning outputs $y_{\text{correct}}$ and incorrect ones $y_{\text{wrong}}$, which are identified using the ground-truth answer $y$. Using these outputs, we construct a preference dataset by distinguishing two cases: (i) when the model generates the correct answer $y_{\text{correct}}$, predicting [eos] is preferred over [refine], and (ii) vice versa for incorrect answers. Concretely, given an input $x$ with its correct reasoning output $y_{\text{correct}}$ and an incorrect output $y_{\text{wrong}}$, we define the preference triplets $(x, y^+, y^-)$ as:

$$\begin{cases} \big(x, \hat{y} \oplus [\texttt{eos}], \hat{y} \oplus [\texttt{refine}]\big), & \text{if } \hat{y} = y_{\text{correct}} \\ \big(x \oplus \hat{y}, [\texttt{refine}], [\texttt{eos}]\big), & \text{if } \hat{y} = y_{\text{wrong}} \end{cases}$$

where $\oplus$ is the concatenation operator. Based on the proposed collection strategy, we generate a preference dataset $\mathcal{D}_{\text{verify}}$ for training the intrinsic verification of the LLM. To this end, we jointly optimize the supervised fine-tuning loss with the direct preference optimization (DPO; Rafailov et al., 2023) loss. Specifically, for a given preference dataset $\mathcal{D}$, the SFT and DPO preference losses are defined as:

$$\mathcal{L}_{\text{SFT}}(\mathcal{D}) := -\mathbb{E}_{(x, y^+) \sim \mathcal{D}} \log \mathcal{M}(y^+ | x)$$

$$\mathcal{L}_{\text{Pref}}(\mathcal{D}) := -\mathbb{E}_{(x, y^+, y^-) \sim \mathcal{D}} \Big[ \sigma \big( r(x, y^+) - r(x, y^-) \big) \Big]$$

$$\text{where} \quad r(x, y) = \beta \log \frac{\mathcal{M}(y \mid x)}{\mathcal{M}_0(y \mid x)},$$

where $\beta \in \mathbb{R}^+$ is hyper-parameter controlling proximity to the base model $\mathcal{M}_0$ and $\sigma$ is the logistic function. It is worth noting that SFT loss only focuses on minimizing the negative log-likelihood of the positive output, i.e., enforcing the model to predict the correct reasoning and answer.

Then, our training objective for self-verification is as:

$$\mathcal{L}_{\text{verify}} := \mathcal{L}_{\text{SFT}}(\mathcal{D}_{\text{verify}}) + \lambda \, \mathcal{L}_{\text{Pref}}(\mathcal{D}_{\text{verify}}) \quad (1)$$

where $\lambda \in \mathbb{R}^+$ is a loss balancing hyperparameter. Here, we denote the initial model $\mathcal{M}_0$ trained with $\mathcal{L}_{\text{verify}}$ as $\mathcal{M}_1$, which is the output model of first curricula.

**Stage 2: Learning to correct self-generations.** We now describe how to train ReVISE to acquire another core ability: self-correction. Similar to self-verification, we perform preference learning using the same loss function. To this end, we aim to construct a new preference dataset, denoted as $\mathcal{D}_{\text{correct}}$. The core idea consists of two main components. First, the curriculum learning: we utilize outputs generated by the model $\mathcal{M}_1$ and initialize stage 2 training from $\mathcal{M}_1$. Second, to learn how to correct incorrect outputs, we repurpose the wrong reasoning paths $y_{\text{wrong}}$ used in stage 1 to construct the dataset.

Concretely, we consider two possible cases: whether the initial response is correct $y_{\text{correct}}$ or incorrect $y_{\text{wrong}}$. If the initial response is correct $y_{\text{correct}}$, we construct preference

data as same as stage 1, i.e., discouraging the generation of [refine] and encouraging [eos]. The key case is when the initial response is incorrect $y_{\text{wrong}}$. In this case, we need to have a positive preference sample that refines the incorrect reasoning $y_{\text{wrong}}$ with the correct reasoning. To achieve this, we concatenate the ground-truth label $y$ to the response. Formally, the preference pairs are defined as:

$$\begin{cases} \big(x, \hat{y} \oplus [\texttt{eos}], \hat{y} \oplus [\texttt{refine}]\big), & \text{if } \hat{y} = y_{\text{correct}} \\ \big(x \oplus \hat{y}, [\texttt{refine}] \oplus y, [\texttt{eos}]\big), & \text{if } \hat{y} = y_{\text{wrong}} \end{cases}$$

where $y$ is the ground-truth label. Using the self-correction preference dataset $\mathcal{D}_{\text{correct}}$, we train the final model $\mathcal{M}_2$ from $\mathcal{M}_1$ with the following correction loss:

$$\mathcal{L}_{\text{correct}} := \mathcal{L}_{\text{SFT}}(\mathcal{D}_{\text{correct}}) + \lambda \, \mathcal{L}_{\text{Pref}}(\mathcal{D}_{\text{correct}}). \quad (2)$$

It is worth noting that stage 2 explicitly defines when and how refinements should be applied, preventing overgeneration and improving response accuracy. By distinguishing between necessary and unnecessary refinements, the model ensures efficient self-correction while simulating multi-step reasoning for complex scenarios.

Furthermore, our dataset collection strategy shares similarities with recent backtracking methods in that incorrect initial generations are utilized to create negative pairs (Zhang et al., 2024c). We also observe that leveraging past failure trajectories aids in ultimately achieving successful reasoning. In this regard, we believe that applying ReVISE to safety-critical applications, akin to backtracking, is an interesting future direction, where our proposed curriculum learning and explicit self-verification stage can contribute to developing safer models.

### 3.3. Verification Confidence-Aware Sampling

We propose an inference method for models trained with ReVISE. The key idea is to calibrate the standard sampling-based scoring approach using the self-verification confidence. Specifically, we apply this method to majority voting, where $N$ samples are generated, and the most frequent prediction is selected. Unlike conventional approaches, our method explicitly accesses the self-verification confidence, as our model not only generates an answer but also determines its correctness by producing either an [eos] or [refine] token. This allows us to directly obtain the probability associated with self-verification, enabling confidence-weighted aggregation for more reliable predictions.

Concretely, given an input $x$, we generate $N$ candidate answers $\mathcal{Y} = \{y_1, y_2, \ldots, y_N\}$ from the LLM at stage 2, denoted as $\mathcal{M}$ for simplicity, where each $y_i$ is sampled as $y_i \sim \mathcal{M}(\cdot | x)$. To refine the selection process, we leverage the softmax probability of the verification (i.e., the probability of [eos] token), denoted as follows:

$$c_i = \mathcal{M}([\texttt{eos}] | y_i, x),$$

*Table 1.* Accuracy (%) for ReVISE (Ours) and other baselines, including Few-shot CoT, SFT, RFT, STAR$^+$ trained models. We consider two math reasoning benchmarks, GSM8K (Cobbe et al., 2021) and MATH-500 (Lightman et al., 2023). MATH-500 is a subset of the original MATH benchmark (Hendrycks et al., 2021). Maj@K indicates that majority voting for K samples, exceptionally ReVISE used its own verification confidence-aware majority voting. The **bold** indicated the best result within the group.

| | Llama-3.2-1B | | | | Llama-3.1-8B | | | |
| | GSM8K | | MATH-500 | | GSM8K | | MATH-500 | |
| Methods | Maj@1 | Maj@5 | Maj@1 | Maj@5 | Maj@1 | Maj@5 | Maj@1 | Maj@5 |
|---|---|---|---|---|---|---|---|---|
| Few-shot CoT | 5.7 | 7.2 | 3.0 | 3.2 | 56.7 | 58.3 | 23.4 | 23.2 |
| SFT (Brown et al., 2020a) | 22.1 | 26.4 | 10.4 | 11.4 | 58.2 | 64.8 | 27.8 | 33.2 |
| RFT (Yuan et al., 2023) | 26.2 | 28.6 | 12.6 | 12.8 | 58.9 | 65.3 | 30.8 | 35.6 |
| STaR$^+$ (Zelikman et al., 2022) | 26.2 | 29.9 | 11.4 | 13.4 | 59.2 | 64.9 | 30.4 | 32.8 |
| **ReVISE (Ours)** | **28.1** | **32.8** | **13.4** | **14.8** | **61.6** | **69.2** | **33.6** | **37.6** |

as a confidence score. Instead of selecting the most frequent prediction, we accumulate these scores by summing the confidence values of identical answers, leading to the final prediction as follows:

$$y^* = \arg\max_{y \in \mathcal{Y}} \sum_{i:y_i=y} c_i.$$

This approach calibrates the traditional majority voting method by weighting predictions based on their model-derived confidence, showing effective scaling at test time.

## 4. Experiments

We provide an empirical evaluation of ReVISE by investigating the following questions:

- Can ReVISE enhance reasoning performance? (Table 1)

- Does confidence-aware sampling improve the performance? (Figure 2 and Figure 6)

- Does/How does the proposed curriculum learning improve the performance? (Figure 3)

- Can ReVISE perform self-verification and -refinement? (Figure 5 and Figure 7)

**Training setup.** For the main experiment, we train ReVISE on Llama-3 models with 1B and 8B parameters, which are not instruction-tuned. We avoid using instruction-tuned models to prevent potential bias from exposure to the gold data of the tasks (Wang et al., 2024). For this reason, the models were first supervised fine-tuned using the labeled dataset, followed by fine-tuning with each respective method. For GSM8K (Cobbe et al., 2021), we train ReVISE using the original training split. For MATH (Hendrycks et al., 2021), we train ReVISE using a 50k subset of Meta-Math (Yu et al., 2024), an augmented version of MATH, and use a 3k subset for the validation set, respectively. Here, MetaMath was employed to mitigate the performance degradation caused by the limited size of the original MATH.

**Baselines.** We compare our method against several baseline approaches: Supervised Fine-Tuning (SFT), RFT(Yuan et al., 2023), and STaR$^+$. In RFT, fine-tuning is performed on supervised fine-tuning data $\mathcal{D}$ and correctly generated samples selected from $k$ completions for each input in the training set by a tuned model. Like RFT, STaR (Zelikman et al., 2022) trains on correctly generated samples, including self-generated rationales given a hint (rationalization). However, unlike RFT, STaR iteratively refines this process without relying on $\mathcal{D}$. Since both ReVISE and RFT utilize ground truth data $\mathcal{D}$, we introduce an extended version of STaR that incorporates SFT data as a baseline, referred to as STaR$^+$. Essentially, STaR$^+$ functions as a multi-iteration variant of RFT with rationalization. We run STaR$^+$ for three iterations, sampling $k$ completions per iteration (GSM8K: $k = 10$, MATH: $k = 4$, GSM240K: $k = 1$) with a temperature of 0.7 for both RFT and STaR$^+$. We initialize a model for each iteration from $\mathcal{M}_0$ that is supervised fine-tuned with $\mathcal{D}$ at each iteration for STaR$^+$ to prevent overfitting.

**Evaluation setup.** We mainly report Majority Voting at K (Maj@K) as a sampling-based metric, exceptionally ReVISE used verification confidence-aware majority voting as described in Section 3.3 (unless otherwise specified). We evaluate ReVISE and baselines on GSM8K (Cobbe et al., 2021) and MATH-500 (Hendrycks et al., 2021), a widely used evaluation benchmark subset of MATH.

### 4.1. Main Results

We first present the main result by comparing the math problem-solving performance with other baselines. Here, we mainly compare ReVISE with various fine-tuning schemes that use a single network and do not use reinforcement learning. Furthermore, we demonstrate the performance of each method with simple test-time scaling methods (i.e., majority voting for baseline methods and using our verification-aware sampling for ReVISE). Also, we verify that ReVISE effectively enhances reasoning in the coding

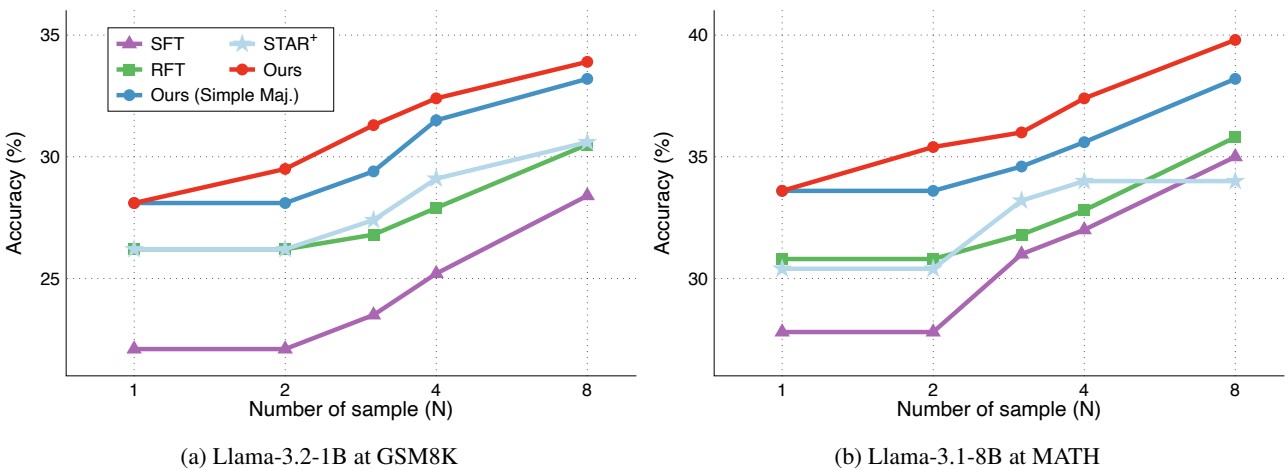

(a) Llama-3.2-1B at GSM8K

(b) Llama-3.1-8B at MATH

*Figure 2.* Test-time scaling comparison between ReVISE (Ours) and baselines, including SFT, RFT, STAR$^+$, and majority voting for ReVISE (Ours (Simple Maj.)) at sampling sizes $N \in \{1, 2, 3, 4, 8\}$. (a) Results for Llama-3.2-1B on the GSM8K dataset. (b) Results for Llama-3.2-8B on the MATH dataset. ReVISE consistently outperforms baselines across all sample sizes and datasets.

domain (i.e., MBPP (Austin et al., 2021)).

As shown in Table 1, we present the math-solving performance of ReVISE compared to other baselines. Overall, ReVISE significantly and consistently outperforms all prior baseline methods. It is worth noting that for both GSM8K and MATH-500, ReVISE achieves the highest Maj@1, indicating that ReVISE is already strong without the proposed sampling scheme. For instance, ReVISE attains 33.6% for Maj@1, significantly outperforming SFT (30.4%) and few-shot CoT (23.4%) on MATH-500 with Llama-3.1-8B. In addition, with the proposed confidence-aware majority voting, ReVISE marked a 4.0% gain after refinement and consistently outperforms other baselines under five sampled answers. These results demonstrate that ReVISE enhances problem-solving accuracy and improves test-time scaling abilities.

*Table 2.* Results on the MBPP (Austin et al., 2021) benchmark for ReVISE and baselines trained on Llama-3.2-1B.

| Method | Pass@1 |
|---|---|
| Few-shot CoT | 24.5 |
| SFT | 30.0 |
| RFT | 29.6 |
| STaR$^+$ | 30.7 |
| **ReVISE (Ours)** | **33.1** |

As shown in Table 2, we further investigate the performance of ReVISE in the coding benchmark MBPP (Austin et al., 2021). Specifically, ReVISE surpasses all baseline approaches, achieving a Pass@1 score of 33.1%, notably outperforming strong baselines such as SFT (30.0%) and STaR$^+$ (30.7%). These results highlight ReVISE's effectiveness beyond mathematical reasoning, extending its superior refinement capabilities to code-generation tasks as well. The consistent performance improvement across diverse bench-

marks underscores the generalizability and robustness of the intrinsic refinement strategy employed by ReVISE.

### 4.2. Inference Scalability of ReVISE

In this section, we evaluate the inference scalability of ReVISE. To this end, we visualize how the test-time scaling improves as one samples more candidates. Specifically, we conduct experiments using our method with different sample sizes $N \in \{2, 3, 4, 8\}$ and compare with results of other baselines using majority voting. As shown in Figure 2, ReVISE achieves significant and consistent gain in all setups. For instance, ReVISE shows a large gap with the strongest baseline RFT, showing 3.3% of improvement in MATH-500 at $N = 8$. Furthermore, our method even benefits under limited number of samples ($N = 2$), while majority voting does not show improvement. This is because majority voting does not use confidence and, hence, can not benefit from small samples (e.g., if all predictions are disjoint, the majority voting does not work). Finally, ReVISE shows scalable improvements in all model configurations, ranging from relatively small 1B models to large 8B models. Notably, ReVISE achieves a significant performance gain in the 8B model, suggesting strong generalization capabilities.

### 4.3. Additional Analysis and Ablation

In this section, we provide a detailed analysis of ReVISE to validate the effect of each proposed component. Unless otherwise specified, we use a Llama-3.2-1B trained on GSM8K across all methods throughout this section.

**Effectiveness of curriculum learning.** We validate the effectiveness of the proposed curriculum learning (in Figure 3). To this end, we train two types of models. First, the model trains without curriculum by optimizing SFT $\mathcal{L}_{\text{SFT}}$

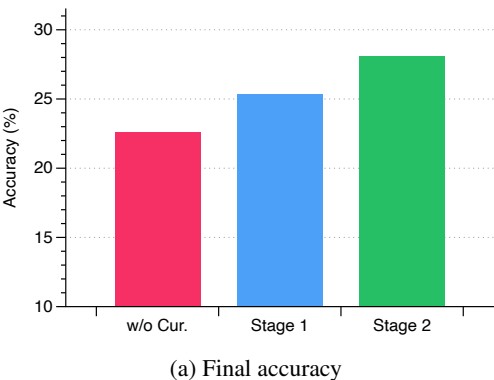

(a) Final accuracy

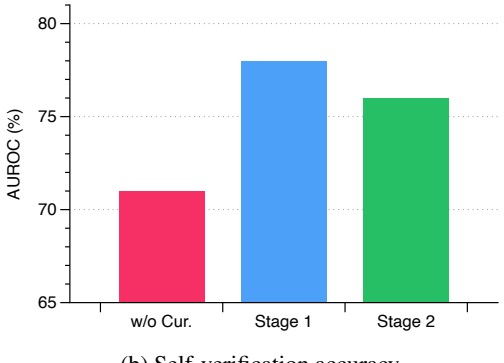

(b) Self-verification accuracy

*Figure 3.* Ablation study on curriculum learning in the aspect of (a) final accuracy (%) and (b) self-verification accuracy reported with AUROC (%). The experiments are conducted using Llama-3.1-1B on the GSM8K dataset. The comparison includes a model trained without curriculum learning (w/o Cur.), trained for only stage 1 (Stage 1), and trained using the full two-stage curriculum learning approach (ReVISE) (Stage 2). (a) Accuracy improves with curriculum learning by mitigating conflicts between competing objectives during early training stages. (b) AUROC results demonstrate enhanced classification performance of corrected and incorrect responses and effective transfer from Stage 1 to the final ReVISE model.

and preference $\mathcal{L}_{\texttt{Pref}}$ loss by using all preference dataset at once, i.e., $\mathcal{D}_{\texttt{correct}}$. Second, we train the model only using the first verification loss, i.e., $\mathcal{D}_{\texttt{verify}}$ (note that self-verification already enables the model to generate the answer but does not know how to correct the generation). As shown in Figure 3a, the curriculum is indeed showing a significant improvement over no curriculum baseline (even the model has used the same preference dataset); two-stage curricula improve the performance from 22.6% to 28.1%.

To further investigate this phenomenon, we evaluate the self-verification accuracy of each method, which measures the model's ability to predict whether its own output is correct. In Figure 3b, we report the verification accuracy in terms of the Area Under the Receiver Operating Characteristic Curve (AUROC) for three models. Notably, the model without curriculum learning achieves an AUROC of 71%, while two-stage curriculum learning improves this to 76%. This suggests that curriculum learning enhances self-verification, allowing the model to refine its predictions based on more reliable verification signals. However, we observe that training at stage 2 slightly degrades verification accuracy, indicating that the self-correction task $\mathcal{D}_{\texttt{correct}}$ is particularly challenging and may lead to catastrophic forgetting (McCloskey & Cohen, 1989). Exploring optimization strategies that improve self-verification and self-correction without compromising overall performance remains an interesting direction for future work.

**Effectiveness of preference learning.** The role of DPO loss in ReVISE is to guide the model to prefer refining when the initial attempt is incorrect and terminating otherwise. Additionally, in our DPO objective, we applied SFT loss to the chosen sequence as introduced in Liu et al. (2024) which applied SFT loss to the selected sequence, $\mathcal{L}_{\texttt{Ours}} := \mathcal{L}_{\text{SFT}}(\mathcal{D}) + \lambda \, \mathcal{L}_{\texttt{Pref}}(\mathcal{D})$, where $\lambda$ is a constant. Specifically,

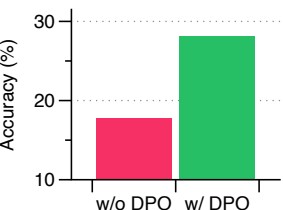

*Figure 4.* Ablation study on DPO loss, evaluated on the GSM8K benchmark. Removing DPO loss significantly reduces accuracy.

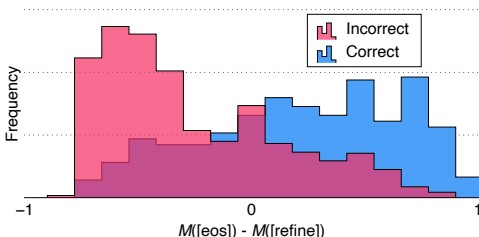

*Figure 5.* Distribution histogram of $\mathcal{M}([\texttt{eos}]) - \mathcal{M}([\texttt{refine}])$ (ignored context $x$ for simplicity). $\mathcal{M}([\texttt{eos}]) - \mathcal{M}([\texttt{refine}]) = 0$ is the threshold of ReVISE trigger intrinsically refine or not. Experiments are conducted using the Llama-3.2-1B model.

ablation experiments without the DPO loss—where only the SFT loss is utilized—in Figure 4 show that ReVISE without DPO demonstrates significantly lower performance $-10.3\%$ compared to the full-trained ReVISE. This indicates that the DPO loss is critical in ReVISE for effectively guiding the refinement process.

**Analysis on the self-verification confidence of ReVISE.** We further analyze the confidence distribution in self-verification to assess whether the model's confidence is well aligned with actual correctness. To this end, we visualize the probability gap between [eos] and [refine] for a given context $x$, simply defined as $\mathcal{M}([\texttt{eos}]) - \mathcal{M}([\texttt{refine}])$. As

*Table 3.* Results on the GSM8K benchmark for ReVISE and baselines trained on Llama-3.2-1B Instruct. Except for ReVISE, all methods underperform compared to the zero-shot CoT baselines.

| Methods | GSM8K | GSM240K |
|---|---|---|
| Zero-shot CoT | 48.6 | 48.6 |
| SFT | 41.9 | 54.8 |
| RFT | 44.0 | 50.9 |
| **ReVISE (Ours)** | **52.3** | **59.4** |

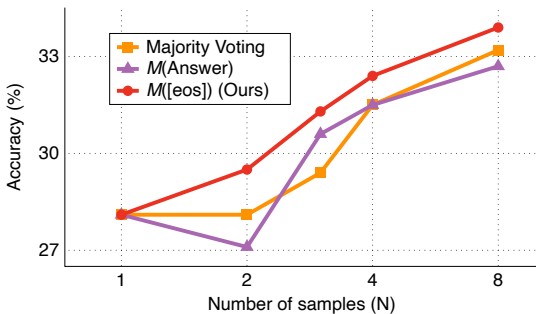

*Figure 6.* Inference-time scaling comparison between ReVISE ($\mathcal{M}([\texttt{eos}])$ (Ours)) and inference metrics. For $\mathcal{M}(\text{Answer})$ and $\mathcal{M}([\texttt{eos}])$ (Ours), we have done weighted majority voting. ReVISE consistently outperforms other inference metrics, and ReVISE's sampling method using weighted majority voting exceeds the performance of majority voting. Experiments are conducted using the Llama-3.2-1B model.

shown in Figure 5, incorrect responses tend to have lower [eos] probabilities, whereas correct responses exhibit higher [eos] probabilities. This demonstrates the model's intrinsic ability to assess its own correctness. Moreover, these results suggest that confidence serves as a reliable metric for calibrating the sampling score, further validating the effectiveness of our confidence-aware sampling method.

**ReVISE on instruction-tuned models.** While we have primally focused on pretrained models and initialized $\mathcal{M}_0$ with the given supervised fine-tuning dataset $\mathcal{D}$ due to the possible data contamination, we also have conducted an experiment on Llama-3.2-1B-Instruct, i.e., instruction-tuned model. Interestingly, as shown in Table 3, all fine-tuning methods, except for ReVISE, underperform the zero-shot CoT baseline when training with GSM8K. This outcome aligns with the widely recognized challenge that fine-tuning instruction-tuned models often leads to catastrophic forgetting, hindering their ability to learn new information effectively with a small-sized dataset. Meanwhile, ReVISE remains notably resistant to this issue. We hypothesize that this advantage stems from how ReVISE utilizes the gold label $y$—only incorporating it as a revised second-attempt completion rather than directly fine-tuning it. In contrast, baselines such as SFT, RFT, and STaR$^+$ rely on fine-tuning the base model on $\mathcal{D}$, which becomes problematic when the target model's performance is already strong, as it struggles to gain further improvements from $\mathcal{D}$.

To this end, we also trained the model using GSM240K, a subset of MetaMath dataset (Yu et al., 2024), which expands the original data about 30-fold by rephrasing questions and answers. As shown in Table 3, while training with GSM240K improved the performance of the SFT baseline, ReVISE still exhibited better performance. This result suggests that ReVISE can adapt to various data characteristics, even in heavily augmented settings.

**Ablation study on confidence-aware sampling.** We explore the impact of different score calibrations during inference by leveraging ReVISE's self-verification mechanism to enable test-time compute-scalable inference strategies (see Section 3.3). Specifically, we compare three scoring schemes: (1) weighted majority voting using $\mathcal{M}([\texttt{eos}]|x)$, (2) unweighted majority voting, and (3) scoring based on

the model's predicted answer likelihood. These calibration methods govern both the selection of candidate answers and the evaluation of their validity.

As shown in Figure 6, $\mathcal{M}([\texttt{eos}]|x)$-based (Ours) score consistently outperforms alternatives across GSM8K benchmarks. For example, with eight sampled candidates, $\mathcal{M}([\texttt{eos}]|x)$-based scoring achieves an accuracy of 33.9%, compared to 33.2% (unweighted majority), and 32.7% (likelihood-based). The trend persists across all tested sampling budgets, suggesting strong compatibility with self-verification mechanisms. This consistent advantage implies $\mathcal{M}([\texttt{eos}]|x)$ better aligns with the model's intrinsic verification capability to distinguish correct reasoning paths. We carefully hypothesize that $\mathcal{M}([\texttt{eos}]|x)$ acts as a latent indicator of solution correctness, as premature termination often correlates with reasoning errors.

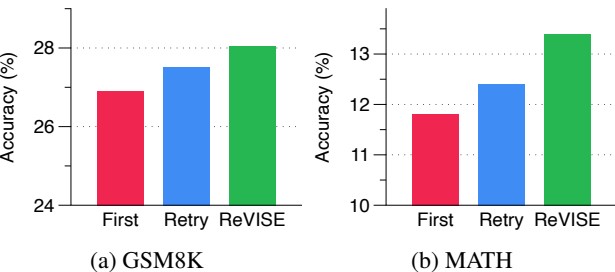

(a) GSM8K      (b) MATH

*Figure 7.* Analysis of refinement capability of ReVISE. We compare the accuracy (%) on GSM8K and MATH when using different decoding approaches. *First* stops at [refine], *Retry* re-generates responses, while ReVISE refines its initial reasoning. The results show that ReVISE improves accuracy, demonstrating its ability to refine rather than randomly re-generate responses. Experiments are conducted using the Llama-3.2-1B model.

**Analysis on the refinement.** We demonstrate that ReVISE refines its answers based on the initial attempt rather than randomly generating a new completion. To evaluate this,

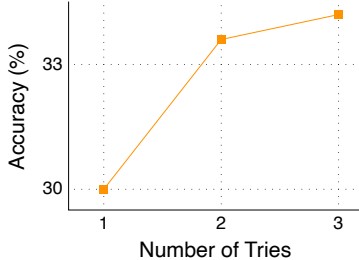
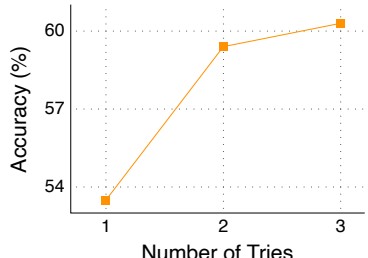

(a) Llama-3.2-8B fine-tuned in MATH and evaluated at MATH-500

(b) Llama-3.2-1B-instruct evaluated at GSM8K

*Figure 8.* Accuracy improvements through iterative refinement. The plot shows the accuracy (%) at GSM8K and MATH-500 of ReVISE across multiple rounds of iterative refinement (1, 2, and 3 tries).

*Table 4.* Accuracy (%) on GSM8K under transfer domain generalization. All models are trained on MATH and evaluated on GSM8K. Trained on Llama-3.2-1B and Llama-3.1-8B models.

| Model | Methods | Accuracy (%) |
|---|---|---|
| Llama-3.2-1B | SFT | 7.3 |
| | RFT | 8.2 |
| | STaR$^+$ | 8.0 |
| | **ReVISE (Ours)** | **8.8** |
| Llama-3.1-8B | SFT | 60.3 |
| | RFT | 60.3 |
| | STaR$^+$ | 58.7 |
| | **ReVISE (Ours)** | **61.5** |

we compare ReVISE with two baselines: *First* and *Retry*. *First* terminates decoding at the [refine] token, while *Retry* generates a new completion upon encountering [refine]. Specifically, *Retry* greedily decodes the first attempt, and if [refine] appears, it samples a new completion with a temperature of $0.7$ following the prompt $x$. In contrast, both *First* and ReVISE greedily generate completions. As shown in Figure 7, ReVISE outperforms both *First* and *Retry*. This result highlights that ReVISE does not generate new responses arbitrarily but instead meaningfully refines and improves upon its initial answer.

**Generalization under transfer dataset domain.** We demonstrate the generalization ablity of ReVISE in a transfer domain setting (see Table 4). Specifically, we train ReVISE on the MATH domain and test on the GSM8K domain, using both Llama-3.2-1B and Llama-3.1-8B models. As shown in Table 4, both model sizes significantly outperform other baselines in this out-of-distribution evaluation. For example, ReVISE achieves an accuracy of 8.8% using Llama-3.2-1B, and 61.5% using Llama-3.1-8 B. These results demonstrate that our method possesses strong domain transferability, maintaining its advantage over baselines even when evaluated on a different dataset.

**Iterative refining sequentially at test time.** Although ReVISE is trained to refine its output in a single pass, we explore its potential for iterative refinement. Specifically, after generating the second attempt, we append it to the original

prompt $x$ and treat it as the first-attempt output. This allows the model to either output [eos] to terminate the sequence or generate a third attempt following the same process, effectively enabling multiple rounds of refinement. As shown in Figure 8, in MATH-500, ReVISE trained on Llama-3.1-8B's accuracy consistently improves as the model iteratively refines its responses up to 3 times.

This observation suggests the potential for training a model explicitly designed for sequential iterative refinement to enhance the iterative refinement even more. By incorporating iterative refinement directly into the training process, the model could learn to self-correct more effectively across multiple rounds. We leave this direction as an exciting avenue for future work.

## 5. Conclusion

In this paper, we introduce Refine via Intrinsic Self-Verification (ReVISE), a novel framework that enables Large Language Models (LLMs) to perform self-verification and self-correction during inference. Through a structured curriculum learning approach, we demonstrated how LLMs can progressively learn to verify their reasoning and improve their outputs. Our results across various reasoning benchmarks show that ReVISE significantly improves first-attempt accuracy while maintaining efficiency. Furthermore, the self-verification mechanism and a confidence-aware decoding strategy enhance model performance without introducing additional computational overhead.

## Impact Statement

This work advances the reasoning of Large Language Models (LLMs) by introducing Refine via Intrinsic Self-Verification (ReVISE), a framework that enables self-verification and self-correction. This has potential applications requiring precise reasoning, such as automated tutoring and decision support systems. Specifically, ReVISE can benefit safety by rigorously revising the model's response in the aspect of the model.

## Acknowledgements

This work was supported by Institute for Information & communications Technology Planning & Evaluation (IITP) grant funded by the Korea government(MSIT) (No.RS-2019-II190075 Artificial Intelligence Graduate School Program(KAIST); No. RS-2024-00509279, Global AI Frontier Lab) and NIPA(National IT Industry Promotion Agency), through the Ministry of Science and ICT (Hyperscale AI flagship project).

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

# A. Experimental Details

In this section, we describe the experimental details of Section 4, including ReVISE and the baselines.

**Dataset details.** In this section, we describe the dataset we used in training and evaluation. Also, explain how we generated the additional datasets.

- **Grade School Math 8K (GSM8K).** The GSM8K dataset (Cobbe et al., 2021) consists of 8,790 high-quality grade-school math word problems. We used the provided train and test splits, ensuring consistency across all experiments. The dataset serves as a benchmark for evaluating the arithmetic and reasoning capabilities of language models.

- **MATH.** The MATH dataset (Hendrycks et al., 2021) is a challenging collection of problems from high school mathematics competitions, covering diverse topics such as algebra, geometry, calculus, and statistics. We utilized the original train and test splits, which include approximately 12,500 problems. Due to the dataset's complexity, it effectively evaluates the model's ability to handle higher-level mathematical reasoning.

- **MetaMath.** MetaMath (Yu et al., 2024) is an augmented version of the MATH and GSM dataset, designed to address the challenges posed by the limited size of the original dataset. We selected a 50k subset of MetaMath for training and sampled 3k problems for the validation set. MetaMath includes additional examples generated using synthetic data augmentation techniques, such as problem paraphrasing and structural variations, to enhance diversity and improve generalization. This augmentation mitigates performance degradation associated with small datasets while maintaining the original problem difficulty and format.

- **MBPP.** MBPP (Austin et al., 2021) is a collection of crowd-sourced Python programming problems. Each instance consists of a natural language task description, a reference solution, and three test cases written in Python. Since ReVISE requires intermediate reasoning steps not provided in the original dataset, we generated them by applying Chain-of-Thought prompting to GPT-4o. For each problem, we collected 16 valid reasoning paths along with corresponding code solutions that successfully pass all test cases.

**Training details of ReVISE** We use AdamW optimizer with a learning rate $\mathtt{lr} \in \{10^{-4}, 10^{-5}\}$ with 10% warm up and cosine decay and train it for one epoch. We trained with batch size 32 for fine-tuning and 64 for preference tuning . For the $\lambda$ constant for SFT loss, we used $\lambda = 0.1$. During the training, for the data sampling phase, we sampled 10 times for each sample in GSM8K and 4 times for each sample in MATH.

**Training model details.** We mainly used the open-source Large Language Models (LLMs) from Llama-family. Specifically we used meta-llama/Llama-3.2-1B, meta-llama/Llama-3.1-8B which are not instruction-tuned and meta-llama/Llama-3.2-1B-Instruct, which is instruction-tuned. We used the model checkpoint from huggingface library.

**Evaluation details.** Used lm-eval-harness for greedy decoding experiments and used our code to evaluate models in sampling settings. Since the output depends on the evaluation batch size, we fixed the batch size to 128 for a fair comparison.

- **GSM8K, MBPP.** Used the test split as a benchmark dataset.

- **MATH-500.** The MATH-500 dataset is a curated collection of 500 MATH dataset. For our experiments, we used Math-500 exclusively for evaluation purposes.

**Resource details.** For the main development we mainly use Intel(R) Xeon(R) Platinum 8480+ CPU @ 790MHz and a 8 NVIDIA H100 GPUs. Additionally, we used NVIDIA RTX4090 GPUs for evaluation.

**Baseline details.**

- **SFT** We fine-tuned the model using a language modeling loss, exploring learning rates from $1e^{-6}$ to $1e^{-4}$, with epochs ranging from 1 to 3 and a batch size of 32.

- **RFT** We sampled ten completions for GSM8K, one for GSM240K, and four for MATH-50K. The model was trained for one epoch on the collected dataset with a fixed learning rate of $1e^{-5}$.

- **STaR$^+$** We sampled the same number of samples as in **RFT**. The outer loop was fixed to 3 for all datasets, with one epoch per outer loop. Rationalization was performed with a hint, where the answer was provided except for the rationale, which served as the hint. The learning rate was fixed at $1e^{-5}$.

# B. Additional Results

## B.1. Comparison with DPO

*Table 5.* Comparison between DPO and ReVISE. We report accuracy (%) on GSM8K and MATH-500. The models are trained on Llama-3.2-1B. The **bold** indicates the best result within the group.

| Method | GSM8K | MATH-500 |
|---|---|---|
| DPO | 22.6 | 10.8 |
| **ReVISE (Ours)** | **28.1** | **13.4** |

To further analyze the effectiveness of ReVISE's training framework, we include a comparison with a reinforcement learning(RL)-based baseline using Direct Preference Optimization (DPO). We trained the DPO model from the same supervised fine-tuned checkpoint, which was used in training ReVISE. We construct a preference pair dataset where ground truth answers are preferred over incorrect responses. As shown in Table 5, ReVISE even outperforms the DPO trained baseline, indicating 22.7% in GSM8K and 10.8% in MATH-500.

## B.2. Iterative Training and Comparison with Self-Correction Work.

*Table 6.* Comparison between ReVISE and other baselines (i.e., Zero-shot CoT, SFT, RFT, and SCoRe (Kumar et al., 2024)). We report accuracy (%) on MATH-500. The models are trained on Gemma-2-2B. The **bold** indicates the best result within the group.

| Method | Accuracy (%) | Training Efficiency |
|---|---|---|
| Zero-shot CoT | 16.8 | - |
| SFT | 17.6 | - |
| RFT | 18.6 | - |
| SCoRe | 23.0 | x1 |
| **ReVISE (Ours)** | 23.2 | **x30** |
| **+ iter1 (Ours)** | 24.2 | x20 |
| **+ iter2 (Ours)** | **25.8** | x15 |

We include a comparison with the self-correction baseline, SCoRe (Kumar et al., 2024). While SCoRe relies on costly online reinforcement learning and requires extensive reasoning path generation (resulting in approximately 1.5 million generations for 3,000 steps with a batch size of 512), ReVISE is significantly more efficient. Specifically, ReVISE constructs preference pairs by generating a single reasoning path per sample, totaling only 50,000 generations for the entire dataset. This corresponds to a 30× reduction in training cost compared to SCoRe. Despite this large efficiency gap, ReVISE achieves higher accuracy than SCoRe on the MATH-500 benchmark using the same Gemma2-2B model, as shown in Table 6. Furthermore, ReVISE 's performance improves further with repeated training cycles. At each cycle, we re-sample reasoning path pairs with the current model and iteratively apply preference optimization to refine it further. This strategy leads to continual accuracy gains while maintaining substantial efficiency benefits. For instance, after two additional iterations, ReVISE achieves 25.8% accuracy, which further increases its performance margin over SCoRe while incurring a training cost that is 15 times smaller.

These results demonstrate that ReVISE is not only more practical and scalable for self-correction tasks, but also able to leverage iterative refinement to reach even higher performance. All comparisons utilize SCoRe's results from the original paper, as there is a lack of open-source code.

## B.3. Quantify the Verifying Performance.

*Table 7.* Comparison of AUROC (%) of verifying correctness between ReVISE and V-STaR (Hosseini et al., 2024) verifier. We report AUROC (%) on GSM8K. The models are trained on Llama-3.2-1B.

| Method | AUROC (%) |
|---|---|
| V-STaR verifier | 69.5 |
| **ReVISE (Ours)** | **76.0** |

We further evaluate the quality of the verifying signal produced by ReVISE using the area under the receiver operating characteristic curve (AUROC) and compare with V-STaR (Hosseini et al., 2024) verifier, while we reported ReVISE's

calibration performance using AUROC in Section 4 and Figure 3b. As shown in Table 7, ReVISE achieves an AUROC of 76.0%, outperforming the V-STaR verifier, which achieves 69.5%, even though V-STaR uses a separately trained verifier. These experimental results demonstrate the effectiveness of ReVISE's intrinsic verifier, resulting in improved performance and enhanced test-time scalability.

### B.4. Extended Test-time Scaling Behavior Experiment.

*Table 8.* Test-time scaling results (Maj@K) on GSM8K with Llama-3.2-1B. We evaluate accuracy (%) as the number of sampled generations K increases from 2 to 64. We compared ReVISE and other baselines (i.e., SFT, RFT, and STaR$^+$).

| | Maj@2 | Maj@4 | Maj@8 | Maj@16 | Maj@32 | Maj@64 |
|---|---|---|---|---|---|---|
| SFT | 20.5% $\pm$ 0.5 | 24.5% $\pm$ 0.6 | 28.2% $\pm$ 0.6 | 30.0% $\pm$ 0.4 | 31.8% $\pm$ 0.1 | 32.1% $\pm$ 0.4 |
| RFT | 24.6% $\pm$ 0.3 | 27.5% $\pm$ 0.3 | 29.8% $\pm$ 0.4 | 30.9% $\pm$ 0.5 | 31.3% $\pm$ 0.3 | 33.2% $\pm$ 0.2 |
| STaR$^+$ | 24.0% $\pm$ 1.1 | 27.1% $\pm$ 0.6 | 29.3% $\pm$ 0.6 | 30.4% $\pm$ 0.6 | 31.1% $\pm$ 0.5 | 31.6% $\pm$ 0.4 |
| **ReVISE (Ours)** | **28.3%** $\pm$ 0.7 | **32.5%** $\pm$ 0.9 | **34.9%** $\pm$ 0.5 | **36.2%** $\pm$ 0.4 | **37.2%** $\pm$ 0.5 | **37.7%** $\pm$ 0.5 |

We extended our experiments on test-time scaling behavior of ReVISE and other baselines (i.e., SFT, RFT, and STaR$^+$) on GSM8K using the Llama-3.2-1B models. Specifically, for the baselines, we measure accuracy by majority voting over K sampled generations, where K ranges from 2 to 64 (Maj@K). For ReVISE, we used our own method, confidence-based weighted majority voting, for the same generation ranges. As shown in Table 8, ReVISE consistently outperforms every baseline across every sample size. Notably, ReVISE reaches 37.7% accuracy at Maj@64, representing a substantial improvement over prior methods. These results demonstrate the effectiveness of ReVISE in leveraging increased generation diversity for robust test-time self-verification.

## B.5. ReVISE Results Example

### B.5.1. REFINEMENT VIA INTRINSIC SELF-VERIFICATION EXAMPLE

*Table 9.* An example of ReVISE's refinement on a GSM8K question

| Question | Mrs. Tatiana owns a grocery store that sells different fruits and vegetables, which includes carrots. The price of carrots in the grocery store increases by 5% of the original price every year. What would be the price of carrots after three years if it was $120 initially? (Round to the nearest integer) Let's think step by step. Put your final answer at the end with 'The answer is: .' |
|---|---|
| Response | The price of carrots increases by 5/100 * $120 = $6. So the price of carrots after one year is 120+6 = $126. The price of carrots after two years is 126+6 = $132. The answer is: $\boxed{132}$ [refine] The price of carrots increases by 5/100*$120 = $6 per year. The price after one year is $120 + $6 = $126 The price after two years is $126 + $6 = $132. The price after three years is $132 + $6 = $138. The answer is: $\boxed{138}$ |
| Target Answer | 138 |

*Table 10.* An example of ReVISE's refinement on a MATH question

| Question | Ten treeks weigh as much as three squigs and one goolee. Two treeks and one goolee are equal in weight to one squig. The combined weight of how many treeks equals the weight of one squig? Let's think step by step. Put your final answer at the end with 'The answer is: .' |
|---|---|
| Response | We can set up a system of equations to solve this problem. Let $t$ be the weight of one treek, $s$ be the weight of one squig, and $g$ be the weight of one goolee. From the given information, we can write the following equations: $10t = 3s + g$  $2t + g = s$ We can solve this system of equations to find the weight of one treek in terms of the weight of one squig. Multiplying the second equation by 3, we get: $6t + 3g = 3s$ Subtracting the first equation from this equation, we get: $6t + 3g - 10t = 3s - (3s + g)$ Simplifying, we get: $-4t + 3g = -g$ Adding $4t$ to both sides, we get: $3g = 4t$ Dividing both sides by 3, we get: $g = \frac{4}{3}t$ Substituting this into the second equation, we get: $2t + \frac{4}{3}t = s$ Multiplying both sides by 3, we get: $6t + 4t = 3s$ Combining like terms, we get: $10t = 3s$ Since we know that $10t = 3s + g$, we can substitute $g = \frac{4}{3}t$ into this equation: $10t = 3s + \frac{4}{3}t$ Multiplying both sides by 3, we get: $30t = 9s + 4t$ Subtracting $4t$ from both sides, we get: $26t = 9s$ Dividing both sides by 9, we get: $\frac{26}{9}t = s$ So, the weight of one treek is equal to $\frac{9}{26}$ times the weight of one squig. Therefore, the combined weight of how many treeks equals the weight of one squig is $\frac{9}{26}$. The answer is: $\boxed{\dfrac{9}{26}}$ [refine] Let $t$, $s$, and $g$ represent the weights of one treek, one squig, and one goolee, respectively. From the given information, we have the following equations: $10t = 3s + g$ (Equation 1) $2t + g = s$ (Equation 2) We want to find the weight of one treek in terms of the weight of one squig, so we need to solve for $t$ in terms of $s$. We can start by solving Equation 2 for $g$ in terms of $t$ and $s$: $g = s - 2t$ Substituting this into Equation 1, we have: $10t = 3s + (s - 2t)$ Simplifying, we get: $10t = 4s - 2t$ Combining like terms, we have: $12t = 4s$ Dividing both sides by 4, we get: $3t = s$ Therefore, the weight of one treek is equal to $\frac{1}{3}$ of the weight of one squig. The answer is: $\boxed{3}$ |
| Target Answer | 3 |

## B.5.2. STOP VIA INTRINSIC SELF-VERIFICATION EXAMPLE

*Table 11.* An example where ReVISE chooses not to refine on a GSM8K question

| | |
|---|---|
| Question | Jen is planning to sell her root crops. She has 6 yams which can be sold at $1.5 each, 10 sweet potatoes that cost $2 each, and 4 carrots which cost $1.25 each. If she sells everything, how much will she earn? |
| Response | Jen earns 6 x $1.5 = $9 for the yams. She earns 10 x $2 = $20 for the sweet potatoes. She earns 4 x $1.25 = $5 for the carrots. Therefore, Jen earns $9 + $20 + $5 = $34 in all. The answer is: $\boxed{34}$ . [eos] |
| Target Answer | 34 |

*Table 12.* An example where ReVISE chooses not to refine on a MATH question

| | |
|---|---|
| Question | Evaluate $(1 + 2i)6 - 3i$. Let's think step by step. Put your final answer at the end with 'The answer is:.' |
| Response | $(1 + 2i)6 - 3i = 6 + 12i - 3i = 6 + 9i$. The answer is: $\boxed{6 + 9i}$ . [eos] |
| Target Answer | $6 + 9i$ |

# C. The few-shot prompt used for evaluation

Tables 13-15 present the few-shot templates used for evaluation. We filled in the {*placeholders*} using using the questions, answers, and test cases (for MBPP only) from Tables 16-18.

*Table 13.* The few-shot template used for GSM8K

| Template for GSM8K |
| --- |
| Given the following problem, reason and give a final answer to the problem.
Problem: {*question*}
Your response should end with "The final answer is [answer]" where [answer] is the response to the problem.
{*answer*}. |

*Table 14.* The few-shot template used for MATH

| Template for MATH-500 |
| --- |
| Problem: {*question*}
Answer: {*answer*}. |

*Table 15.* The few-shot template used for MBPP

| Template for MBPP |
| --- |
| You are given a programming problem. Let's reason step by step before writing the code. Think through the problem carefully, explain your reasoning clearly, and then at the very end, provide your final code.
Here is your task: {*question*}
Your code should pass these tests, and do not include the following test code in your Python code:
{*test cases*}
{*answer*} |

*Table 16.* The 8 few-shot examples used for evaluation on GSM8K

| Question | Answer |
|---|---|
| There are 15 trees in the grove. Grove workers will plant trees in the grove today. After they are done, there will be 21 trees. How many trees did the grove workers plant today? | There are 15 trees originally. Then there were 21 trees after some more were planted. So there must have been 21 - 15 = 6. The final answer is 6 |
| If there are 3 cars in the parking lot and 2 more cars arrive, how many cars are in the parking lot? | There are originally 3 cars. 2 more cars arrive. 3 + 2 = 5. The final answer is 5 |
| Leah had 32 chocolates and her sister had 42. If they ate 35, how many pieces do they have left in total? | Originally, Leah had 32 chocolates. Her sister had 42. So in total they had 32 + 42 = 74. After eating 35, they had 74 - 35 = 39. The final answer is 39 |
| Jason had 20 lollipops. He gave Denny some lollipops. Now Jason has 12 lollipops. How many lollipops did Jason give to Denny? | Jason started with 20 lollipops. Then he had 12 after giving some to Denny. So he gave Denny 20 - 12 = 8. The final answer is 8 |
| Shawn has five toys. For Christmas, he got two toys each from his mom and dad. How many toys does he have now? | Shawn started with 5 toys. If he got 2 toys each from his mom and dad, then that is 4 more toys. 5 + 4 = 9. The final answer is 9 |
| There were nine computers in the server room. Five more computers were installed each day, from mondax"y to thursday. How many computers are now in the server room? | There were originally 9 computers. For each of 4 dayxx's, 5 more computers were added. So 5 * 4 = 20 computers were added. 9 + 20 is 29. The final answer is 29 |
| Michael had 58 golf balls. On tuesday, he lost 23 golf balls. On Wednesday, he lost 2 more. How many golf balls did he have at the end of wednesday? | Michael started with 58 golf balls. After losing 23 on tuesday, he had 58 - 23 = 35. After losing 2 more, he had 35 - 2 = 33 golf balls. The final answer is 33 |
| Olivia has $23. She bought five bagels for $3 each. How much money does she have left? | Olivia had 23 dollars. 5 bagels for 3 dollars each will be 5 x 3 = 15 dollars. So she has 23 - 15 dollars left. 23 - 15 is 8. The final answer is 8 |

*Table 17.* The 4 few-shot examples used for evaluation on MATH

| Question | Answer |
|---|---|
| Find the domain of the expression $\frac{\sqrt{x-2}}{\sqrt{5-x}}$. | The expressions inside each square root must be non-negative. Therefore, $x - 2 \geq 0$, so $x \geq 2$, and $5 - x \geq 0$, so $x \leq 5$. Also, the denominator cannot be equal to zero, so $5 - x > 0$, which gives $x < 5$. Therefore, the domain of the expression is $\boxed{[2, 5)}$. The answer is $\boxed{[2, 5)}$. |
| If $\det \mathbf{A} = 2$ and $\det \mathbf{B} = 12$, then find $\det(\mathbf{AB})$. | We have that $\det(\mathbf{AB}) = (\det \mathbf{A})(\det \mathbf{B}) = (2)(12) = \boxed{24}$. The answer is $\boxed{24}$. |
| Terrell usually lifts two 20-pound weights 12 times. If he uses two 15-pound weights instead, how many times must Terrell lift them in order to lift the same total weight? | If Terrell lifts two 20-pound weights 12 times, he lifts a total of $2 \cdot 12 \cdot 20 = 480$ pounds of weight. If he lifts two 15-pound weights instead for $n$ times, he will lift a total of $2 \cdot 15 \cdot n = 30n$ pounds of weight. Equating this to 480 pounds, we can solve for $n$: $$30n = 480 \Rightarrow \quad n = 480/30 = \boxed{16}$$ The answer is $\boxed{16}$. |
| If the system of equations $$6x - 4y = a, 6y - 9x = b,$$ has a solution $(x, y)$ where $x$ and $y$ are both nonzero, find $\frac{a}{b}$, assuming $b$ is nonzero. | If we multiply the first equation by $-\frac{3}{2}$, we obtain $$6y - 9x = -\frac{3}{2}a.$$ Since we also know that $6y - 9x = b$, we have $$-\frac{3}{2}a = b \Rightarrow \frac{a}{b} = \boxed{-\frac{2}{3}}.$$ The answer is $\boxed{-\frac{2}{3}}$. |

*Table 18.* The 3 few-shot examples used for evaluation on MBPP

| Question | Test Cases | Answer |
|---|---|---|
| Write a function to find the similar elements from the given two tuple lists. | • assert similar_elements((3, 4, 5, 6),(5, 7, 4, 10)) == (4, 5)
• assert similar_elements((1, 2, 3, 4),(5, 4, 3, 7)) == (3, 4)
• assert similar_elements((11, 12, 14, 13),(17, 15, 14, 13)) == (13, 14) | Step 1: Analyze the Problem
We are given two tuples of integers and asked to find the "similar elements" between them. This implies we need to identify the elements that are common to both tuples.
The result should be returned as a tuple, and the order of elements in the result should be sorted in ascending order, based on the test cases.
Step 2: Plan the Solution
  1. **Input:** Two tuples, e.g., '(3, 4, 5, 6)' and '(5, 7, 4, 10)'.
  2. **Goal:** Find common elements between the two tuples.
  3. **Output:** A tuple of the common elements in sorted order.
Steps:
  • Convert both tuples into sets. This allows us to perform set intersection efficiently.
  • Use the '&' operator or '.intersection()' method to get common elements.
  • Convert the resulting set to a sorted list.
  • Convert that sorted list back into a tuple.
  • Return the result.
Step 3: Consider Edge Cases
  • If there are no common elements, the result should be an empty tuple: '()'.
  • The order of elements in the result must be sorted ascendingly.
Now that the reasoning is clear, let's move on to the code.
The answer is:

```python
def similar_elements(t1, t2):
    return tuple(sorted(set(t1) & set(t2)))
``` |
| Write a function to identify non-prime numbers. | • assert is_not_prime(2) == False
• assert is_not_prime(10) == True
• assert is_not_prime(35) == True | Step 1: Analyze the Problem
We are asked to write a function 'is_not_prime(n)' that determines whether a number is **not** a prime. The function should return:
  • 'True' if the number is **not** a prime.
  • 'False' if the number **is** a prime.
Step 2: Understand the Requirements
  • A **prime number** is a number greater than 1 that has no divisors other than 1 and itself.
  • So, the function should:
    – Return 'True' if 'n' is less than 2 (since 0 and 1 are not prime).
    – Return 'True' if 'n' has any divisor other than 1 and itself.
    – Return 'False' otherwise.
Step 3: Plan the Solution
  1. Handle edge cases: If 'n <= 1', return 'True' because these are not prime.
  2. Loop from '2' to 'sqrt(n)': - If any number in this range divides 'n', it is not prime → return 'True'.
  3. If the loop completes without finding a divisor, return 'False'.
Step 4: Write and Test the Code
The answer is:

```python
import math

def is_not_prime(n):
    if n <= 1:
        return True
    for i in range(2, int(math.sqrt(n)) + 1):
        if n % i == 0:
            return True
    return False
``` |
| Write a function to find the largest integers from a given list of numbers using heap queue algorithm. | • assert heap_queue_largest( [25, 35, 22, 85, 14, 65, 75, 22, 58],3)==[85, 75, 65]
• assert heap_queue_largest( [25, 35, 22, 85, 14, 65, 75, 22, 58],2)==[85, 75]
• assert heap_queue_largest( [25, 35, 22, 85, 14, 65, 75, 22, 58],5)==[85, 75, 65, 58, 35] | Step 1: Analyze the Problem
We are given a list of integers and a number 'n', and we need to find the 'n' largest integers from that list using a *heap queue algorithm*.
This means:
  • We should use a *heap*, which is a special kind of binary tree used primarily for priority queues.
  • Python provides a built-in module called 'heapq' which supports heap operations.
Key point:
  • Although 'heapq' is a min-heap by default, it provides utility functions to get the 'n' largest elements efficiently: 'heapq.nlargest(n, iterable)'.
Step 2: Plan the Solution
  1. Import the 'heapq' module.
  2. Use 'heapq.nlargest(n, iterable)' to get the largest 'n' elements from the list.
  3. Return that list.
Step 3: Write the Code
This will be a single function 'heap_queue_largest(nums, n)' that uses 'heapq.nlargest'.
The answer is:

```python
import heapq

def heap_queue_largest(nums, n):
    return heapq.nlargest(n, nums)
``` |

