# OpenReview forum: "ReVISE: Learning to Refine at Test-Time via Intrinsic Self-Verification"
_ICML.cc/2025/Conference — ICML 2025 poster_

### Official Review · Reviewer_nE5C · 2025-02-24

**Overall Recommendation:** 3

**Summary:**

This paper introduces Refine via Intrinsic Self-Verification (ReVISE), an efficient framework that enables large language models (LLMs) to self-correct their outputs through self-verification. ReVISE allows LLMs to evaluate their reasoning processes and iteratively refine their outputs based on verification feedback. To achieve this efficiently, the framework employs a structured curriculum leveraging online preference learning. Given the dual challenges of self-verification and reasoning correction, ReVISE applies curriculum learning to address them sequentially, gathering both failed and successful reasoning paths to construct preference pairs for efficient training. During inference, ReVISE naturally scales by integrating self-verification and correction, further enhanced by a confidence-aware decoding mechanism. Experimental results across various reasoning tasks show that ReVISE enables efficient self-correction and significantly enhances LLM reasoning performance.

## update after rebuttal
The authors' rebuttal addressed my concerns, leading me to revise my recommendation from weak reject to weak accept.

**Claims And Evidence:**

Some claims are not well supported by evidence:
1. In Section 4.3, it mentions that “We observe that training at stage 2 slightly degrades verification accuracy, indicating that the self-correction task is particularly challenging and may lead to catastrophic forgetting”. I think the results cannot demonstrate that the self-correction task is particularly challenging and may lead to catastrophic forgetting. Because in Figure 3a, the accuracy after stage 2 is improved significantly compared to that of stage 1.

2. In Section 4.3, it states that the results in Figure 5 suggest that confidence serves as a reliable metric for calibrating the sampling score. However, from the figure, we can see that there are some correct cases that have low confidence and some incorrect cases that have high confidence. So the calibration is not very good. I think it needs to provide a quantitative metric like TPR at 5% FPR to quantify the calibration performance and compare the proposed method with current state-of-the-art baselines.

**Essential References Not Discussed:**

No.

**Experimental Designs Or Analyses:**

Yes, I check the soundness of the experimental designs and analyses in Section 4. I have some concerns:

1. It only considers Llama-3 models with 1B and 8B parameters. It would be better to evaluate other kinds of models (e.g. Mistral models).
2. It seems the instruction-tuned models have much better performance on GSM8K and MATH-500. It would be better to compare the proposed method to those instruction-tuned models.
3. I think it needs to compare the proposed method with the SCoRe method proposed in [1].

[1] Kumar, A., Zhuang, V., Agarwal, R., Su, Y., Co-Reyes, J. D., Singh, A., Baumli, K., Iqbal, S., Bishop, C., Roelofs, R., Zhang, L. M., McKinney, K., Shrivastava, D., Paduraru, C., Tucker, G., Precup, D., Behbahani, F., and Faust, A. Training language models to self-correct via reinforcement learning. arXiv preprint arXiv:, 2024.

**Methods And Evaluation Criteria:**

The proposed method makes sense for the problem. However, the evaluation criteria have some limitations:
1. It only reports Majority Voting at N where N is up to 8. It should consider larger N (e.g. N=64).
2. It only evaluates the proposed method on math benchmarks. The paper would be stronger if it considers other kinds of datasets like reasoning and coding datasets (e.g. LiveBench and LiveCodeBench).

**Other Comments Or Suggestions:**

No other comments.

**Other Strengths And Weaknesses:**

Strengths:
1. The paper is well-written, with a thorough discussion of related work.
2. The proposed test-time sampling scheme, which integrates self-verification confidence, is novel and enhances output reliability.

Weaknesses:
1. The proposed method only employs single-round self-correction. It would be better to investigate whether multi-round self-correction can further improve performance.
2. Running experiments multiple times and reporting the mean and variance of the results would strengthen the evaluation.

**Questions For Authors:**

I can raise my scores if the authors clarify the following concerns:
1. Would the proposed method benefit from multi-round self-correction?
2. Could the authors compare the proposed method to the SCoRe method?
3. Could the authors provide additional evidence to support the claim that the proposed confidence serves as a reliable metric for calibrating the sampling score?

**Relation To Broader Scientific Literature:**

The key contributions of this paper align with the recent trend of training LLMs to improve self-correction [1], as well as leveraging self-verification to enhance inference-time scaling efficiency [2].

[1] Kumar, A., Zhuang, V., Agarwal, R., Su, Y., Co-Reyes, J. D., Singh, A., Baumli, K., Iqbal, S., Bishop, C., Roelofs, R., Zhang, L. M., McKinney, K., Shrivastava, D., Paduraru, C., Tucker, G., Precup, D., Behbahani, F., and Faust, A. Training language models to self-correct via reinforcement learning. arXiv preprint arXiv:, 2024.

[2] Snell, C., Lee, J., Xu, K., and Kumar, A. Scaling llm testtime compute optimally can be more effective than scaling model parameters. arXiv preprint arXiv:2408.03314, 2024.

**Theoretical Claims:**

No theories.

---

> ### Author Rebuttal · Authors · 2025-04-01
>
> Dear reviewer nE5C,
>
> We sincerely appreciate your efforts and comments to improve the manuscript. We respond to your comment in what follows.
>
> ---
>
> **[Q1] Would the proposed method benefit from multi-round self-correction?**
>
> We have already investigated multi-round self-correction in Appendix B.1 of our paper. Specifically, after generating the first refined response ($y_2$), we generate a second refined response ($y_3$) by appending ($y_2$) to the original prompt ($x$). Formally, $y_2 \sim p(\cdot \mid x, y_1), y_3 \sim p(\cdot \mid x, y_2)$.
>
> As shown in Figure 8, ReVISE improves with each refinement round in MATH-500, e.g., 1st: 33.0 %, 2nd: 33.6 %, 3rd round: 34.2 %, demonstrating the effectiveness of multi-round self-correction. We will move the multi-round results to the main part of the final version.
>
> ---
>
> **[Q2] Compare with SCoRe method.**
>
> We first note that our primary focus was to build an efficient way to implement self-correction, where SCoRe requires a heavy online RL; it needs to generate the reasoning path for every training mini-batch, generating 1.5M paths in total (512 batch size × 3,000 steps). In contrast, ReVISE only requires a single reasoning path generation per individual sample to construct a preference pair, resulting in 50k generations. Hence, we expect that our training cost is 30 (=1.5M / 50k) times smaller than SCoRe.
>
> Nevertheless, we agree that SCoRe is a worthwhile method to compare, and compared with ReVISE on Gemma2 2B. Here, we have reported SCoRe’s results from the paper (as it is not open-sourced and difficult to reproduce with our computing resources). As shown in the table below, ReVISE outperforms SCoRe on MATH using the same Gemma 2 2B model. We thank the reviewer for the suggestion and will include the comparison in the final version.
>
> Table: https://bit.ly/4cdMnqN
>
> ---
>
> **[Q3] Could the authors provide additional evidence to support the claim that the proposed confidence serves as a reliable metric for calibrating the sampling score?**
>
> We remark that we have already reported the quantified ReVISE’s calibration performance using AUROC in Section 4.3 and Figure 3 (b). To further support this claim, we compared ReVISE with V-Star’s verifier [1]. Notably, while ReVISE relies on an intrinsic verifier, it still outperforms V-Star’s separately trained verifier, as shown in the table below, with a higher AUROC reflecting superior confidence-correctness alignment.
>
> Table: https://bit.ly/4hW5HtE
>
> [1] Hosseini, Arian, et al. "V-star: Training verifiers for self-taught reasoners." arXiv preprint arXiv:2402.06457 (2024).
>
> ---
>
> **Whether the results in Figure 3a demonstrate that the self-correction task is particularly challenging and may lead to catastrophic forgetting.**
>
> We agree that our original phrasing may have been misleading. We intended to convey that self-correction introduces additional difficulty, as the model must both verify and refine its output—potentially making it harder to maintain verification performance during training.
> We will revise the sentence to avoid strong terms like “catastrophic forgetting” and clarify the trade-off between verification and correction in joint training.
>
> ---
>
> **Benchmark on coding domain.**
>
> Due to space limitations, we kindly refer the reviewer to our response to cTTG’s comment W2, which addresses this point in detail.
>
> ---
>
> **ReVISE to instruction-tuned models.**
>
> We remark that Table 2 already reports results on instruction-tuned models, where ReVISE consistently outperforms other baselines. This shows that ReVISE complements instruction tuning and can further boost strong models.
>
> ---
>
> **Sampling for N=64**
>
> We extended our sampling setup on N=2 to 64. As shown in the table, ReVISE consistently outperforms baselines across all sample sizes. We will include these results in the final version.
>
> Table: https://bit.ly/4hSZ23t
>
> ---
>
> **Running experiments multiple times and reporting the mean and variance.**
>
> As shown in the table above, we report the mean, standard deviation over 5 random seeds for all sample sizes N=2 to 64. ReVISE not only consistently outperforms baseline methods in terms of mean accuracy, but also maintains stable variance. We will report the mean variance across multiple runs in the final version.
>
> ---
>
> **Evaluate other kinds of model**
>
> To address your concern, we conducted experiments on the Gemma2 2B. As shown in the table below, ReVISE still consistently outperforms other baselines, including SFT, RFT, and SCoRe, showing its effectiveness beyond the Llama family. We will include this result in the final version of the paper.
>
> Table: https://bit.ly/4cdMnqN

---

> > ### Comment · Reviewer_nE5C · 2025-04-03
> >
> > After reading the rebuttal, I still have a concern that the proposed method may not significantly outperform the SCoRe method. Thus, I keep my original scores.

---

> > > ### Author Response · Authors · 2025-04-04
> > >
> > > Thank you for your prompt and clear response,
> > > Reviewer nE5C.
> > >
> > > We remark again that ReVISE is **30× more training-efficient** than SCoRe while still achieving higher accuracy. Namely, the training efficiency of ReVISE is its important advantage. However, if one wishes to pay more training cost, ReVISE’s accuracy can be further improved. To be specific, one can re-sample reasoning path pairs using a progressively improved model from the previous iteration, and re-train ReVISE using them by applying DPO [1] iteratively. The results are reported in the table below, which demonstrate that **ReVISE outperforms SCoRe by 2.8% higher accuracy** in testing with **15× speed-up in training**. We think that this is not a marginal improvement. We will update all results of ReVISE in the final draft by applying such an iterative DPO strategy, which we believe further strengthens our paper.
> > >
> > > \begin{array}{clcc}
> > > \hline
> > > \textbf{Test Dataset} & \textbf{Method} & \textbf{Accuracy} & \textbf{Training Efficiency} \newline
> > > \hline
> > > & \text{base} & 16.8    & -    \newline
> > > &\text{SCoRe}  & 23.0   & \times 1    \newline
> > > \text{MATH-500}&\textbf{ReVISE (ours)}   & \textbf{23.2} & \times\textbf{30} \newline
> > > &\textbf{+ iter1 (ours)}    & \textbf{24.2} & \times\textbf{20} \newline
> > > &\textbf{+ iter2 (ours)}    & \textbf{25.8} & \times\textbf{15} \newline
> > > \hline
> > > \end{array}
> > >
> > > Finally, we hope your understanding that SCoRe has not provided open source code (the results of SCoRe in the above table are from the original paper), and we could not perform more extensive comparisons. Since we will publicly release our code and reproducible supplementaries, we think that this would be very useful for follow-up researchers. We think this is also our important contribution to the community. Thank you again for your valuable feedback and consideration.
> > >
> > > [1] Rafailov, Rafael, et al. "Direct preference optimization: Your language model is secretly a reward model." Advances in Neural Information Processing Systems 36 (2023)

---

### Official Review · Reviewer_Aquu · 2025-03-13

**Overall Recommendation:** 2

**Summary:**

The paper introduces DPO to fine-tune LLMs in two steps, which are self-verification and self-correction, and a special token [refine] is introduced. Simple and effective method.

**Claims And Evidence:**

All claims made in the submission supported by clear and convincing evidence.

**Essential References Not Discussed:**

No other references

**Experimental Designs Or Analyses:**

1. The lack of OOD evaluations.
2. The lack of evaluation of LLaMA-8B on GSM8K.

**Methods And Evaluation Criteria:**

The evaluation dataset lacks OOD tests and it is unclear how well the fine-tuned model generalizes.

**Other Comments Or Suggestions:**

No others.

**Other Strengths And Weaknesses:**

strength:

1. The proposed method is simple and effective.

2. The paper integrates self-verification and self-correction with DPO.

Weakness:

1. The datasets employed and experimental designs are not sufficient; the paper proposes to make demonstrations in mathematics and coding domains in lines 81-82, but the actual experiments are only conducted in mathematics.

2. Without OOD validation, the generalization of LLMs after DPO may be somewhat compromised.

3. The concerns of paper on open source.

4. Does the multiple sampling process change the temperature? Given a question, what is the probability that the model always answer correctly?

**Questions For Authors:**

Please see weakness.

**Relation To Broader Scientific Literature:**

The introduction of special token [refine] is a novel idea.

**Theoretical Claims:**

No Theoretical Claims.

---

> ### Author Rebuttal · Authors · 2025-04-01
>
> Dear reviewer Aquu,
>
> We sincerely appreciate your efforts and comments to improve the manuscript. We respond to your comment in what follows.
>
> ---
>
> **[W1] The datasets employed and experimental designs are not sufficient; the paper proposes to make demonstrations in mathematics and coding domains in lines 81-82, but the actual experiments are only conducted in mathematics.**
>
> In response to the concern about limited coverage of task domains, we conducted additional experiments in the coding domain using the MBPP benchmark, as initially proposed in lines 81–82. ReVISE, trained on the Llama 3.2-1B model, achieves 33.1%, outperforming the second-best baseline at 30.7%. Since MBPP lacks reasoning annotations, we generated ground truth reasoning using GPT-4o. These results complement our mathematics experiments and demonstrate that ReVISE generalizes effectively beyond mathematics tasks.
>
> \begin{array}{lc}
> \hline
> \textbf{Method} & \textbf{Pass@1} \newline
> \hline
> \text{3 Shots} & 24.5 \newline
> \text{SFT} & 30.0 \newline
> \text{RFT} & 29.6 \newline
> \text{STaR}^+ & 30.7 \newline
> \text{ReVISE (Ours) } &  \textbf{33.1} \newline
> \hline
> \end{array}
>
> ---
>
> **[W2] The lack of OOD evaluations.**
>
> To address your concern and demonstrate robustness to domain shifts, we conducted an OOD evaluation by training ReVISE on the MATH dataset and evaluating it on GSM8K. As shown in the table below (Maj@1), ReVISE consistently outperforms other baseline methods (the gap decreases a bit though), including SFT, RFT, and StaR+, under the OOD setting. We will add the OOD results to the final draft.
>
> \begin{array}{lcc}
> \hline
> \textbf{Model} & \textbf{Method} & \textbf{GSM8K Acc.} \newline
> \hline
> \text{} & \text{SFT} & 7.3 \newline
> \text{Llama 3.2-1B} & \text{RFT} & 8.2 \newline
> \text{} & \text{STaR}^+ & 8.0 \newline
> \text{} & \text{ReVISE (Ours) } &  \textbf{8.8} \newline \hline
> \text{} & \text{SFT} & 60.3 \newline
> \text{Llama 3.1-8B} & \text{RFT} & 60.3 \newline
> \text{} & \text{STaR}^+ & 58.7 \newline
> \text{} & \text{ReVISE (Ours) } &  \textbf{61.5} \newline
> \hline
> \end{array}
>
> ---
>
> **[W3] The concerns of the paper on open source.**
>
> In the final draft, we will release the training and evaluation code for ReVISE, including all hyperparameter configurations, to ensure full reproducibility of the results presented in the paper.
>
> ---
>
> **[W4] Concern about temperature consistency in the experiment.**
>
> We clarify that we have used a consistent temperature value (t=0.7) throughout the entire experiment. We will include the information about temperature value in the final version of the paper.
>
> ---
>
> **[W5] The lack of evaluation of Llama 3.1-8B on GSM8K.**
>
> Following your suggestion, we have additionally trained and evaluated ReVISE using the Llama 3.1-8B on GSM8K. As shown in the table below (Maj@1), ReVISE consistently outperforms baseline methods under this larger model setting, demonstrating that our approach scales well with model size.
>
> \begin{array}{lc}
> \hline
> \textbf{Method} & \textbf{Accuracy} \newline
> \hline
> \text{SFT} & 58.2 \newline
> \text{RFT} & 58.9 \newline
> \text{STaR}^+ & 59.2 \newline
> \text{ReVISE (Ours) } &  \textbf{61.6} \newline
> \hline
> \end{array}

---

### Official Review · Reviewer_cTTG · 2025-03-14

**Overall Recommendation:** 3

**Summary:**

This paper proposes a method to perform self-correction intrincically with two steps. First is a self-verifying stage, by leveraging SFT and DPO, the LLM will learn how to distiguish between correct reasoning and wrong reasoning. Second step is self-correction, when the LLM generates the refine token, then the LLM needs to refine its generation. The pipeline requires preference datasets to train and it can be served as the test-time refinement during inference stage. The proposed method is a natural way to decompose the refine process.

**Claims And Evidence:**

The experiments in this paper clearly supports that the proposed method is a clear and effective way to improve reasoning performance during inference stage. And the proposed method is also efficient because only incorrect responce needs to refine.

**Essential References Not Discussed:**

N/A

**Experimental Designs Or Analyses:**

The experiments are conducted on math datasets such as GSM8K and MATH-500. Except from performance on benchmarks, this paper also provides several ablation studies to verify the effectiveness of the proposed method and the necessity of several components of the pipeline. The overall performance shows that, in reasoning tasks (math problems in this case), ReVISE achieved comparable improvement over SFT, RFT and STaR+. And the proposed two stage training is essential for the refinement performance.

One issue is that, even though in curriculum learning experiments, the authors shows that by optimizing verification and refinement loss together, the overall performance is weak, compared with proposed method. Since this model leverages reinforcement learning (specifically, DPO), it would be better to compare reinforcement learning setting in main experiment section. That is to say, without this two-stage refinement, how much performance gain can be achieved by RL finetuning (DPO, GRPO, etc.) only?

**Methods And Evaluation Criteria:**

The diversity of the benchmarks is limited, only math datasets are used in experiments. It would be better to increase the range of task types here to fully verify the effectiveness of the proposed method.

**Other Comments Or Suggestions:**

Typo: The caption in Figure 8b should be Llama-3.2-8B rather than Llama-3.2-1B

**Other Strengths And Weaknesses:**

N/A

**Questions For Authors:**

The proposed method can only decide whether the reasoning trajectory is correct or not in the end of responce. Do you think this method can be extended to perform refinement within the reasoning trajectory? For example, when the model generate half of the response, it can realize that the current genereated text is not good enough. In tasks like math, for human, if the current reasoning trajectory has confliction, it's better to go back now and find the nearest position and minimal changes to give a better trajectory. So the second question raises, could this method be elastically extended as a "eraser"? Which means the verification model can detect which part in previous generation should be rewrite? That is to say, the model can perform self-correct in half-way during inference stage, worked in an intervention way.

**Relation To Broader Scientific Literature:**

N/A

**Theoretical Claims:**

N/A

---

> ### Author Rebuttal · Authors · 2025-04-01
>
> Dear reviewer cTTG,
>
> We sincerely appreciate your efforts and comments to improve the manuscript. We respond to your comment in what follows.
>
> ---
>
> **[W1] The diversity of the benchmarks is limited. It would be better to increase the range of task types here to fully verify the effectiveness of the proposed method.**
>
> To address the concern about benchmark diversity, we added experiments on the MBPP coding benchmark. ReVISE, trained on the Llama 3.2-1B model, achieves 33.1%, outperforming the second-best baseline at 30.7%. Since MBPP lacks reasoning annotations, we generated ground truth using GPT-4o, which is used to train the baselines and ReVISE. These results further support ReVISE general applicability across domains.
>
> \begin{array}{lc}
> \hline
> \textbf{Method} & \textbf{Pass@1} \newline
> \hline
> \text{3 Shots} & 24.5 \newline
> \text{SFT} & 30.0 \newline
> \text{RFT} & 29.6 \newline
> \text{STaR+} & 30.7 \newline
> \text{ReVISE (Ours)} & \textbf{33.1} \newline
> \hline
> \end{array}
>
> ---
>
> **[W2] As the baseline, how much performance gain can be achieved by RL finetuning only (specifically DPO)?**
>
> Following your suggestion, we have additionally considered an RL-based baseline, specifically a DPO-trained model. To be more specific, we trained the DPO model from an SFT model and constructed a preference pair dataset that paired the ground truth as chosen and the wrong answer as rejected. As shown in the table below (Maj@1), ReVISE even outperforms the RL-based baseline, indicating the 22.7% in gsm8k and 10.8% in MATH-500.
>
> \begin{array}{lcc}
> \hline
> \textbf{Model} & \textbf{GSM8K} & \textbf{MATH} \newline
> \hline
> \text{DPO} & 22.7 & 10.8 \newline
> \text{ReVISE (Ours)} &  \textbf{28.1} &  \textbf{13.4} \newline
> \hline
> \end{array}
>
> ---
>
> **[Q1] Can ReVISE be extended to perform, refinement within the reasoning trajectory?**
>
> Thank you for your great question. It is true that the current version explicitly verifies after generating all reasoning traces and answers. We believe it will be a great future work if the model can detect errors during the reasoning trace generation, e.g., train the model to emit a [refine] token midway when there is an error in the reasoning trace where the error can be detected by using an LLM judge. We thank the reviewer for the question and will discuss this future direction in the final draft.

---

> > ### Comment · Reviewer_cTTG · 2025-04-05
> >
> > Thank authors for the added experiment results. I will maintain my original positive score on this submission.

---

> > > ### Author Response · Authors · 2025-04-07
> > >
> > > Dear Reviewer cTTG,
> > >
> > > Thank you for letting us know! We are delighted to hear that our rebuttal addressed your questions well.
> > > We will add the additional experimental results and address the suggested question about future direction in the final draft.
> > > If you have any further questions or suggestions, please do not hesitate to let us know.
> > >
> > > Thank you very much,
> > >
> > > Authors

---

### Decision · Program_Chairs · 2025-05-01

**Decision:**

Accept (poster)

**Comment:**

The paper introduces a method for LLMs to self-correct their outputs through self-verification.

It integrates self-verification and self-correction with DPO (Direct Preference Optimization).  The paper is well-written and thoroughly discusses related work. The proposed test-time sampling scheme incorporates self-verification confidence and improves output reliability. The method demonstrates efficient self-correction and improves reasoning performance.

The evaluation datasets primarily focused on math problems but the authors added coding tasks during the rebuttal.
The authors also added new experiments to assess OOD performance.

Overall, we recommend weak acceptance.